# Crosstalk between Mesenchymal Stem Cells and Cancer Stem Cells Reveals a Novel Stemness-Related Signature to Predict Prognosis and Immunotherapy Responses for Bladder Cancer Patients

**DOI:** 10.3390/ijms24054760

**Published:** 2023-03-01

**Authors:** Lin Ma, Hualin Chen, Wenjie Yang, Zhigang Ji

**Affiliations:** Department of Urology, Peking Union Medical College Hospital, Chinese Academy of Medical Science and Peking Union Medical College, Beijing 100000, China

**Keywords:** bladder cancer, cancer stem cell, immunotherapeutic response, mesenchymal stem cell, stemness, intercellular communication, SLC2A3, tumor microenvironment

## Abstract

Mesenchymal stem cells (MSCs) and cancer stem cells (CSCs) maintain bladder cancer (BCa) stemness and facilitate the progression, metastasis, drug resistance, and prognosis. Therefore, we aimed to decipher the communication networks, develop a stemness-related signature (Stem. Sig.), and identify a potential therapeutic target. BCa single-cell RNA-seq datasets (GSE130001 and GSE146137) were used to identify MSCs and CSCs. Pseudotime analysis was performed by Monocle. Stem. Sig. was developed by analyzing the communication network and gene regulatory network (GRN) that were decoded by NicheNet and SCENIC, respectively. The molecular features of the Stem. Sig. were evaluated in TCGA-BLCA and two PD-(L)1 treated datasets (IMvigor210 and Rose2021UC). A prognostic model was constructed based on a 101 machine-learning framework. Functional assays were performed to evaluate the stem traits of the hub gene. Three subpopulations of MSCs and CSCs were first identified. Based on the communication network, the activated regulons were found by GRN and regarded as the Stem. Sig. Following unsupervised clustering, two molecular subclusters were identified and demonstrated distinct cancer stemness, prognosis, immunological TME, and response to immunotherapy. Two PD-(L)1 treated cohorts further validated the performance of Stem. Sig. in prognosis and immunotherapeutic response prediction. A prognostic model was then developed, and a high-risk score indicated a poor prognosis. Finally, the hub gene SLC2A3 was found exclusively upregulated in extracellular matrix-related CSCs, predicting prognosis, and shaping an immunosuppressive tumor microenvironment. Functional assays uncovered the stem traits of SLC2A3 in BCa by tumorsphere formation and western blotting. The Stem. Sig. derived from MSCs and CSCs can predict prognosis and response to immunotherapy for BCa. Besides, SLC2A3 may serve as a promising stemness target facilitating cancer effective management.

## 1. Introduction

Bladder cancer represents one of the most common urological malignancies, with over 500,000 newly diagnosed cases and 200,000 deaths each year worldwide [1]. Bladder cancer can be classified into non-muscle invasive bladder cancer (NMIBC) and muscle-invasive bladder cancer (MIBC) based on the depth of invasion. Although MIBC only accounts for approximately 30% of the newly diagnosed, it is characterized by aggressiveness, metastasis, drug resistance, and recurrence, which are responsible for the decreased cancer-specific survival after R0 resection [2].

Cancer stemness, defined as the stem-cell-like phenotype of cancer cells, including self-renewal and differentiation, plays a critical role in the progression, metastasis, resistance to drugs, and recurrence of several cancers, including colorectal cancer (CRC) [3], hepatocellular carcinoma [4], and BCa [5]. Cancer stem cells (CSCs) and mesenchymal stem cells (MSCs) have been recognized as the main contributors to stemness maintenance [6]. Considering the important roles of CSCs in tumor initiation, conventional drug resistance, and the origin of metastasis, they have been considered the targets in cancer treatment [7]. MSCs infiltrate into the tumor microenvironment (TME) and promote tumor development through the secretion of pro-survival factors. In TME, MSCs support the cancer stemness by protecting tumor cells from physiological stress and therapies [6]. Furthermore, the exosome secretion or extracellular vesicles facilitate the intercellular crosstalk and promote angiogenesis, progression, resistance, and quiescent cancer cell activation [8]. Therefore, decoding the communication networks may shed light on the cancer stemness and identify potential therapeutic targets enhancing effective cancer treatment.

Compared to conventional bulk RNA-seq technology, single-cell RNA-seq (scRNA-seq) facilitates decoding the intricate communication networks and uncovering molecular mechanisms at the single-cell level. In the study, we first integrated two BCa scRNA-seq datasets and built the intercellular communication network between CSCs and MSCs. Then, the gene regulatory network (GRN) analysis was performed to identify activated regulons (the transcription factors and their target genes) within the communication network. A stemness-related signature (Stem. Sig) was subsequently constructed, and it showed predictive values for prognosis and immunotherapy response. Ultimately, we identified the hub gene SLC2A3 of the Stem. Sig and validated its biological features in BCa cells by wet-lab experiments. 

## 2. Results

### 2.1. scRNA-Seq Analysis Unravels the Heterogeneity of CSCs and MSCs

Based on cell markers reported in the literature, we first identified CSCs and MSCs populations from the integrated BCa scRNA-seq datasets [6,9,10,11] (Appendix A). Following the Seurat pipeline, we re-clustered CSCs into three subpopulations and presented the top 10 markers of each subpopulation in Figure 1a. Cluster 0 highly expressed collagen gene family including COL4A1, COL3A1, COL4A2, COL1A1, COL1A2, and COL6A2. Thus, we named Cluster 0 “ECM-related CSCs.” Enrichment analysis revealed that collagen fibril organization, extracellular matrix/structure organization, and ECM-receptor interaction pathway were enriched (Figure 1b). Marker genes of Cluster 1 were enriched in tissue/organ development; thus, Cluster 1 was defined as “quiescent CSCs” (Figure 1c). As for Cluster 2, IL-6, CCL2, and CCL21 were upregulated. And Cluster 2 markers were enriched in immune-related biological progress and pathways, such as interferon-gamma, TNF signaling pathway, and complement and coagulation cascades. So, we named Cluster 2 “immune-related CSCs” (Figure 1d,e). Pseudotime analysis demonstrated that quiescent CSCs were projected onto the root of the developmental trajectory, and ECM-related CSCs and immune-related CSCs were projected onto two branches (Figure 1f). BEAM analysis demonstrated that branch-dependent genes were collagen gene family and immune-related genes. They were responsible for quiescent CSCs’ developmental direction (Figure 1g).

MSCs were also clustered into three subpopulations (Appendix A). Markers of Cluster 0 were mainly enriched in leukocyte recruitment-related biological progress, including leukocyte adhesion to vascular endothelial cells, regulation of leukocyte cell-cell adhesion, and regulation of cellular extravasation. Thus, Cluster 0 was defined as “innate immune-related MSCs” (Appendix A). In the top 10 markers of Cluster 1, COL4A1 was identified. In combination with enrichment analysis, we defined Cluster 1 as “ECM-related MSCs” (Appendix A). According to the upregulated genes and enriched terms, we named Cluster 2 “adaptive immune-related MSCs” (Appendix A and Figure 1h). Pseudotime analysis revealed that immune-related MSCs were projected on the root and developed into ECM-related MSCs with the developmental trajectory (Figure 1i). BEAM demonstrated the expression status of the collagen gene family and immune-related genes changed from the early developmental stage to the top right of the tree through branch point 1 (Figure 1j).

### 2.2. Stem. Sig. Derived from Intercellular Communication Networks

In the MSCs-CSCs communication network, ECM-related MSCs served as the main sender cells. VMF, COL4A1, CTGF, and SERPING1 were highly activated in the top 20 ligands. It has been well-documented that the four ligands were involved in tumor cell proliferation, invasion, and migration of several malignancies, including hepatocellular carcinoma, gastric adenocarcinoma, and BCa [12,13,14] (Figure 2a). 

Based on the 20 top ligands, we further constructed a ligand-target network predicting corresponding targets (Figure 2b). In combination with SCENIC, we obtained the highly activated regulons in the ligand-target network and compiled these genes to a communication signature named Stem. Sig (Figure 2c,d, Appendix A).

### 2.3. Stem. Sig. Stratifies the TME of TCGA-BLCA into Two Molecular Subtypes

Based on Stem. Sig., we clustered the TME of TCGA-BLCA into two molecular subclusters via unsupervised consensus clustering. Both the CDF curves and PAC scores indicated the optimal clustering number was 2 (Figure 3a–c). Patients in cluster 1 had more unfavorable prognoses compared to those in cluster 2 (Figure 3d). Besides, cluster 1 had a significantly higher proportion of high-grade and late-stage BCa patients compared to cluster 2, indicating the association between cluster 1 and the clinical progression of BCa (Figure 3e). 

To verify the distinct stemness features, we compared the mRNAsi index and activities of stemness-related signatures between two clusters. Results showed that cluster 1 was characterized by a high mRNAsi index and highly activated signatures, suggesting the higher cancer stemness of tumors in cluster 1 (Figure 3f,g). Robertson et al. [15] reported three main molecular subtypes of TCGA-BLCA, namely luminal subtypes (further divided into Luminal-papillary, Luminal-Infiltrated, and Luminal), one “Basal/Squamous” subtype, and one “Neuronal” subtype. In our study, cluster 1 had a higher proportion of Basal/Squamous BCa compared to cluster 2 (Figure 3h). Functional enrichment analysis revealed that most hallmark gene sets were also upregulated in cluster 1, including EMT, IL-6/JAK/STAT3 signaling, and INFA signaling (Figure 3i). 

### 2.4. Stem. Sig. Models an Inflamed TME of TCGA-BLCA

Considering the profound mechanisms between stemness and immunity, we next analyzed the immune cell infiltration abundances between two clusters. Results showed that the infiltration levels of 28 immune cell subsets were all higher in cluster 1 than those in cluster 2 (Figure 4a). To verify the robustness, we employed six other TME decoding algorithms, including CIBERSORT, EPIC, MCP-counter, quanTIseq, TIMER, and xCell. Similar results were found (Appendix A). Furthermore, most immunity-related factors, including chemokines, MHC molecules, immunostimulators, and immunoinhibitors, were highly expressed in cluster 1 (Appendix A). These findings indicated that the BCa of cluster 1 was characterized by an inflamed TME. 

The enrichment scores of a seven-step anticancer immunity cycle (Figure 4b) and immunotherapy-predicted pathways (Figure 4c) were also higher in cluster 1 compared to those in cluster 2. Besides, both TCR and BCR evenness were higher in cluster 1 (Figure 4d). Pathway enrichment analysis uncovered that cluster 1 was linked to immunity-related pathways, including antigen processing and presentation, natural killer cell-mediated cytotoxicity, and PD-L1 expression and PD-1 checkpoint pathway in cancer, and other common pathways, including JAK-STAT, NK-kappa B, and PI3K-Akt signaling. 

Given the inflamed TME and potently effective immunity in cluster 1, we hypothesized the effective responses to immunotherapy of BCa patients in this cluster. Thus, we analyzed the expression patterns of four immunotherapeutic predictors, including IFNG, CYT, GEP, and TMB, between two clusters. Figure 4f showed that all four factors were significantly highly expressed in cluster 1. Taken together, BCa of cluster 1 had an inflamed TME, and patients in this cluster may benefit from ICIs therapy compared to those in cluster 2. 

### 2.5. Stem. Sig. Predicts Immunotherapeutic Responses in ICIs-Treated BCa Cohorts

To further unravel the value of the Stem. Sig. in predicting immunotherapeutic responses, we analyzed two ICIs-treated BCa cohorts, IMvigor 210 and Rose2021UC. In the IMvigor 210 cohort, two molecular subclusters were identified based on the Stem. Sig. via consensus clustering (Appendix A). Cluster 1 was related to a more favorable prognosis (Figure 5a), a higher proportion of responders (CR/PR, Figure 5b), and a higher proportion of inflamed TME phenotypes (Figure 5c) compared to cluster 2. Similarly, in the two molecular subclusters of Rose2021UC (Appendix A), cluster 1 was linked to better survival (Figure 5d) and a higher proportion of responders (Figure 5e). Besides, the TMB, a positive predictor of immunotherapy response, was higher in cluster 1 (Figure 5f). 

Consistent with these two ICI-treated cohorts, cluster 1 in TCGA-BLCA was supposed to have more responders to both PD-1 and CTLA-4 inhibitors, as predicted by TIDE and SubMap analyses (Figure 5g,h). 

### 2.6. A Prognostic Model Derived from the Stem. Sig. by an Integrated Machine-Learning Framework

To facilitate translational medicine, we decided to develop a consensus model based on the Stem. Sig. that may be friendly used in clinical settings. First, we selected prognosis-related genes from the Stem. Sig. by univariate cox regression analysis. Then, we developed an integrated machine-learning framework of 101 combinations to select the optimal model with the highest C-index. Finally, the optimal prognostic model was constructed by both-direction StepCox and RSF (Figure 6a). The relative importance of each model gene is illustrated in Figure 6b. Besides, the model demonstrated robust prognostic prediction performance in TCGA-BLCA (Figure 6c), GSE31684 (Figure 6d), GSE13507 (Figure 6e), GSE32548 (Figure 6f), and GSE32894 (Figure 6g). 

### 2.7. SLC2A3 Overexpression Promoted CSC Traits, Which Can Be Suppressed by SLC2A3 Inhibition

As the most significant gene in the prognostic model, SLC2A3 was exclusively upregulated in ECM-related CSCs (Figure 7A) and correlated with the stemness score determined by the enrichment score of the Stem. Sig. (Figure 7B). Survival analysis demonstrated the prognostic value of the gene (Figure 7C). Cancer-immunity cycle represented the biological processes of tumor cell elimination [16]. Figure 7D demonstrated that the overexpression of SLC2A3 was associated with an impaired cancer-immunity cycle. Further investigation uncovered that M2 macrophage polarization factors were also upregulated (Figure 7E). 

All these findings suggested the critical roles of SLC2A3 in the stemness of BCa. We further analyzed the stemness traits of SLC2A3 by tumorsphere formation assay. As shown in Figure 8a, the number and sizes of spheres were promoted in SLC2A3 upregulated cells, which was markedly attenuated by SLC2A3 inhibition. Besides, the expression levels of stem cell markers were evaluated (Figure 8b). Results showed that SLC2A3 affected the CSC traits of BCa cells.

## 3. Discussion

Cancer stemness plays a crucial role in tumor initiation, progression, drug resistance, and metastasis. The reciprocal cell communications between CSCs and other cells, especially MSCs, maintain the stemness. Therefore, deciphering the communication networks can shed light on the molecular mechanisms of stemness and facilitate novel biomarker identification. CSCs have been well studied in BCa and are thought to be responsible for the BCa initiation and maintenance of tumor growth [17]. They also regulate the angiogenesis and metastasis of BCa and are associated with a higher risk of recurrence. At the single-cell level, Wang et al. generated a comprehensive BCa cancer-cell atlas consisting of 54,971 single cells and highlighted the critical roles of the CSC population in recurrent BCa [10]. Similarly, a subpopulation with overexpressed cancer stem cell markers SOX9 was discovered in the single cells derived from one T3-stage MIBC [11]. Further SCENIC analysis of the critical TF regulatory network revealed the preferential upregulation of SOX9 and SOX2 in this subpopulation. And the key roles of SOX2 and SOX9 in the regulation of cancer stemness and tumor metastasis have been well-documented in previous studies [18,19]. For MSCs, the biological roles of exosomes in BCa cells have been reported [20,21]. However, the understanding of MSCs at the single-cell level is limited in BCa. In the study, we first identified subpopulations of CSCs and MSCs in the integrated scRNA-seq dataset. Based on the communication network and GRN, the Stem. Sig was developed and showed satisfactory performance in the prediction of prognosis and response to immunotherapy. Finally, a prognostic model involving SLC2A3 was constructed and demonstrated robust performance in prognostic prediction. 

According to the enrichment analysis, the MSCs and CSCs can be categorized into two functional properties: ECM-related and immune-related. ECM serves as a major structural component of the TME and constantly undergoes remodeling progress with tumor development. As an essential role in the stem cell niche, ECM participates in stemness maintenance, stem cell proliferation, and self-renewal [22]. Compared to normal ECM, tumor ECM is characterized by stiffness due to overexpressed collagens. Stiffed ECM comprises a physical barrier that hinders the transport of drugs to the stem cell niche and survives the cancer stem cell. Furthermore, ECM influences the infiltration of immune cells into the TME. ECM promotes the recruitment of M2 macrophages and Tregs, whereas it inhibits the infiltration of CD8+ T cells. For example, the PI3K-AKT signaling pathway was upregulated in our ECM-related MSCs, which facilitates the immune escape of CSCs [23]. Besides, driven by immunomodulatory genes, CSCs reduce the infiltration density of anti-tumor immune cells and sculpt an immunosuppressive TME with a high abundance of pro-tumor immune cells like M2 macrophages. Targeting the cancer stemness will facilitate the chemotherapeutic drug delivery to the stem cell niche and reshape the immunological feature of TME. 

In the communication pattern, ECM-related MSCs functioned as the main sender cells in the ligand-receptor network, with upregulated VEGFC, ADAM17, VWF, and EDN1. As a key regulator of angiogenesis in cancer, VEGFC can be activated by oncogenes, growth factors, and stress, such as hypoxia. Apart from effects on vascular functions, including vascular constriction and normalization, VEGF can promote tumor growth and metastasis by binding receptors on tumor cells and inhibiting the maturation of immune cells. Moreover, VEGF-mediated signaling contributes to the function of CSCs and promotes tumor initiation. The crucial role of VEGF in the tumor niche makes it a promising target for anti-cancer therapy. Previous studies have reported that patients with advanced-stage cancers benefit from VEGF-targeted therapy with or without chemotherapy [24]. The pro-tumoral properties of ADAM17, VWF, and EDN1 have been well-studied in the literature, and targeted therapy has demonstrated the effects of tumor-inhibiting [24,25]. CTGF, COL4A1, and TFGB1 were overexpressed in ECM-related CSCs/MSCs according to the communication pattern. The three genes drive tumorigenesis, invasiveness, and chemotherapeutic resistance in various cancers. Except for COL4A1, the crucial roles of CTGF and TFGB1 in stemness regulation have been previously reported [26,27]. 

Regulons EGR1, MEF2C, and KFL9 were activated in ECM-/immune-related CSCs. For quiescent CSCs, three regulons (GATA3, KLF5, and E2F3) were upregulated, and GATA3 demonstrated exclusively activated status. Chen et al. uncovered the important function of GATA3 in quiescent cellular status. Upregulated GATA3 suggested a dormant status, and GATA3 knockdown induced a proliferative status shift [28]. These activated regulons formed a novel Stem. Sig that divided the bulk tumor samples into two molecular subtypes. Two molecular subtypes with distinct stemness properties were characterized by different immunological phenotypes: high stemness features indicated inflamed TME. Our results were consistent with the previous findings that CSC with high stemness demonstrated unfavorable prognosis, inflamed TME, and high response rate to immunotherapy [3]. 

Based on the Stem. Sig, we developed a prognostic model. Patients in the high-risk score group suffered from unfavorable prognoses. Within the risk model, the hub gene SLC2A3 harbored the highest hazard risk and demonstrated a positive correlation with the Stem. Sig enrichment score. In our study, upregulated SLC2A3 contributed to the unfavorable prognosis of BCa. Similarly, Yang et al. developed a prognostic signature including SLC2A3 for BCa patients and found that overexpressed SLC2A3 was related to high-risk score (poor prognosis) [29]. The correlation between upregulated SLC2A3 and decreased OS has been widely reported in other solid tumors, including CRC and breast cancer [30,31]. Decoding the molecular mechanisms unraveled that SLC2A3 promoted invasion, EMT progress, and stemness [32]. In BCa, SLC2A3 suppression inhibited tumor cell glucose uptake and proliferation and promoted cell apoptosis [33]. In our study, SLC2A3 was involved in the infiltration of M2 macrophages in BCa and contributed to the impaired anti-tumor immunity cycle, consistent with the findings of a gastric cancer study [34]. Intriguingly, SLC2A3 was found exclusively upregulated in ECM-related CSCs that facilitated ECM stiffness. In addition to the role of SLC2A3 in stem traits of BCa, targeting SLC2A3 may enhance effective cancer treatment. Interestingly, Xu and colleagues 2015 reported the striking findings of targeting SLC2A3 by siRNA-based nanomedicine in glioma therapy [35]. Collectively, targeting SLC2A3 therapy is promising for BCa patients.

## 4. Materials and Methods

### 4.1. Data acquisition and Pre-Processing

Two BCa scRNA-seq datasets were downloaded from the Gene Expression Omnibus (GEO) database by accession number: GSE130001 [36] and GSE146137 (mice data was discarded) [37]. We performed the quality control progress as previously described [38]. The normalization, integration, dimension reduction, and clustering were conducted stepwise according to the Seurat manual [39]. Subsequently, we identified the MSCs and CSCs populations by previously reported cell markers.

BCa bulk RNA-seq datasets were procured from GEO and TCGA databases with the following accession number: TCGA-BLCA, GSE31684, GSE13507, GSE32548, and GSE32894. Two PD-(L)1 treatment datasets, IMvigor210 [40] and Rose2021UC [41], were also obtained.

### 4.2. Subpopulation Identification and Pseudotime Analysis

MSCs and CSCs populations were first extracted from the integrated scRNA-seq dataset and further clustered into subpopulations. We then used the FindAllMarkers function in Seurat to identify positive markers of each subpopulation. By clusterProfiler, enrichment analyses, including Gene Ontology (GO) and Kyoto Encyclopedia of Genes and Genomes (KEGG), were performed to facilitate subpopulation annotation [42]. We further performed pseudotime analysis and built the single-cell developmental trajectory by Monocle [43]. Once the branch point has been selected, the BEAM function in Monocle was used to identify genes that differ between branches or change expression status with the developmental trajectory.

### 4.3. Intercellular Crosstalk, GRN, and Stem. Sig.

To decipher the intricate communication networks, we used the NicheNet to identify putative ligands and binding targets [44]. Top ligands and targets in the communication network were regarded as the communication pattern. SCENIC was subsequently used to identify activated regulons within the pattern [45]. Finally, the Stem. Sig was constructed.

### 4.4. Unsupervised Consensus Clustering, Enrichment Analysis, and Immunotherapy Prediction

After removing normal samples, we identified molecular subtypes of TCGA-BLCA based on the Stem. Sig. by ConsensusClusterPlus package [46]. CDF curves and PAC scores were employed to determine the optimal clustering number. To validate the stemness between subtypes, we obtained 26 stemness-related gene sets from a web-based tool: StemChecker (http://stemchecker.sysbiolab.eu/, accessed on 9 January 2023) [47], and calculated the stemness enrichment scores of each TCGA-BLCA via GSVA [48]. Besides, messenger RNA expression-based stemness index (mRNAsi) was procured from the study by Malta et al. and used to explore the differences between clusters [49].

GSEA enrichment analysis with hallmark gene sets downloaded from Molecular Signatures Database (https://www.gsea-msigdb.org/gsea/msigdb/, accessed on 9 January 2023) was performed to investigate the biological features. Additionally, immune cell infiltration levels were evaluated by CIBERSORT, EPIC, MCP-counter, quanTIseq, TIMER, and xCell, which were implemented in the IOBR R package [50].

We used the Tumor Immune Dysfunction and Exclusion (TIDE) [51] and SubMap analysis [52] to assess the response to immunotherapy. Two anti-PD-(L)1 treated cohorts were analyzed to further evaluate the performance of the Stem. Sig. in predicting immunotherapy response.

### 4.5. Prognostic Model Construction

We first performed a univariate Cox regression analysis to identify prognosis-related genes (*p* < 0.05) based on the Stem. Sig. Then, an integrated machine-learning framework was developed to establish a consensus prognostic model based on several BCa RNA-seq cohorts, including TCGA-BLCA, GSE31684, GSE13507, GSE32548, and GSE32894. To be specific, the framework was developed from 101 combinations of 10 machine-learning algorithms via 10-fold cross-validation, including survival support vector machine (survival-SVM), random survival forest (RSF), elastic network (Enet), generalized boosted regression modeling (GBM), supervised principal components (SuperPC), partial least squares regression for Cox (plsRcox), CoxBoost, stepwise Cox, Ridge, and Lasso. TCGA-BLCA was utilized for training the model, and other cohorts were used to test the performance.

### 4.6. Cell Culture and Transfection

We obtained human bladder cancer cell lines T24 and 5637 from the Cancer Institute of the Chinese Academy of Medical Sciences. The cell line was cultured in Dulbecco’s modified Eagle’s medium (DMEM), supplemented with 10% fetal bovine serum (FBS) and 1% penicillin-streptomycin (Gibco; Thermo Fisher Scientific, Inc., Shanghai, China). Cell lines were grown at 37 °C in a humidified atmosphere of 95% air and 5% CO_2_.

We purchased pcDNA3.1/SLC2A3 (negative control: pcDNA3.1) and small interfering RNA (siRNA) targeting SLC2A3 (si-SLC2A3; negative control: si-NC) from RiboBio (Guangzhou, China). Following the manufacturer’s guidelines, cells were transfected using Lipofectamine 3000 (Invitrogen, Waltham, MA, USA).

### 4.7. Western Blotting and Antibodies

The bladder cancer cells were lysed in RIPA lysis buffer. Protein concentration was measured by a BCA assay kit (Beyotime, Shanghai, China). Protein lysates were separated using 10% SDS-PAGE and transferred onto PVDF membranes. The membranes were blocked with 5% skimmed milk for 1 h at room temperature and then incubated with primary antibody overnight at 4 °C. Following this, the membranes were incubated with the secondary antibody at RT for 1 h. Each blot was detected by an ECL kit.

Primary antibodies used: anti-SLC2A3, anti-SOX2, anti-YAP1, and anti-GAPDH. All antibodies were purchased from Sigma-Aldrich (St. Louis, MO, USA).

### 4.8. Tumorsphere Formation Assay

Bladder cancer cells were seeded into an ultralow-attachment 6-well plate at a density of 5000 cells per well containing DMEM/F12 medium with bFGF (20 ng/mL), EGF (20 ng/mL), insulin (5 µg/mL) and 2% B27 (Gibco; Thermo Fisher Scientific, Inc.). After being cultured with 5% CO_2_ at 37 °C for 7 days, tumorspheres were observed under the inverted microscope.

### 4.9. Statistical Analyses

R software (v 4.1.1) and GraphPad Prism (v 8.0.2) were used to perform all statistical analyses. The Wilcoxon test was used to analyze the differences between the 2 groups. Chi-squared test was used to examine the differences between categorical variables. Pearson correlation coefficient was used for correlation analysis. Kaplan–Meier curves with the log-rank test were performed for survival analysis. A 2-tailed *p*-value < 0.05 was regarded as statistically significant.

## 5. Conclusions

Deciphering crosstalk between MSCs and CSCs identified a Stem. Sig. that predicted prognosis and the response to immunotherapy for BCa. SLC2A3 was exclusively upregulated in ECM-related CSCs and contributed to the stem traits of BCa. Targeting SLC2A3 may facilitate effective cancer management.

## Figures and Tables

**Figure 1 ijms-24-04760-f001:**
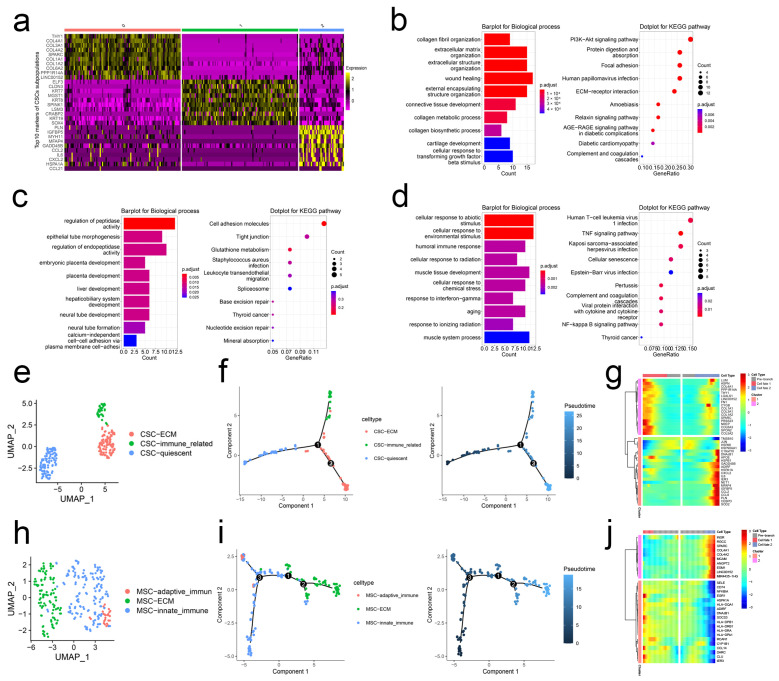
scRNA-seq analysis unravels the heterogeneity of CSCs and MSCs. (**a**) Top 10 markers of each subtype of CSCs. Top 10 terms of BP of GO and KEGG pathway enrichment analysis of Cluster 0 (**b**), 1 (**c**), and 2 (**d**). (**e**) Subpopulations of CSCs. (**f**) Pseudotime and developmental trajectory of the subpopulation of CSCs. (**g**) BEAM of MSCs. (**h**) Subpopulations of MSCs. (**i**) Pseudotime and developmental trajectory of the subpopulation of MSCs. (**j**) BEAM of MSCs.

**Figure 2 ijms-24-04760-f002:**
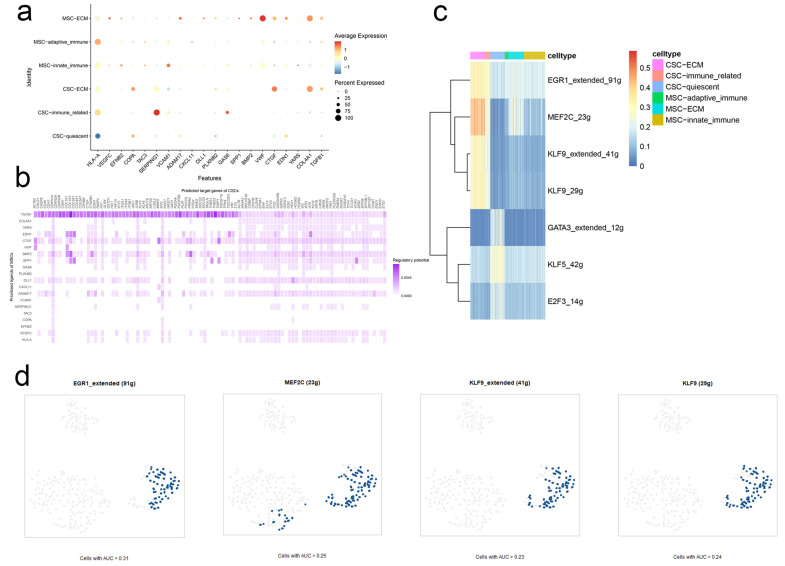
Stem. Sig. derived from intercellular communication networks. (**a**) The expression profiles of the top 20 ligands in the communication network. (**b**) The ligand-receptor network was constructed by NicheNet. (**c**) Activated regulons identified by SCENIC. (**d**) Regulons EGR1, MEF2C, and KLF9 were highly activated in CSCs.

**Figure 3 ijms-24-04760-f003:**
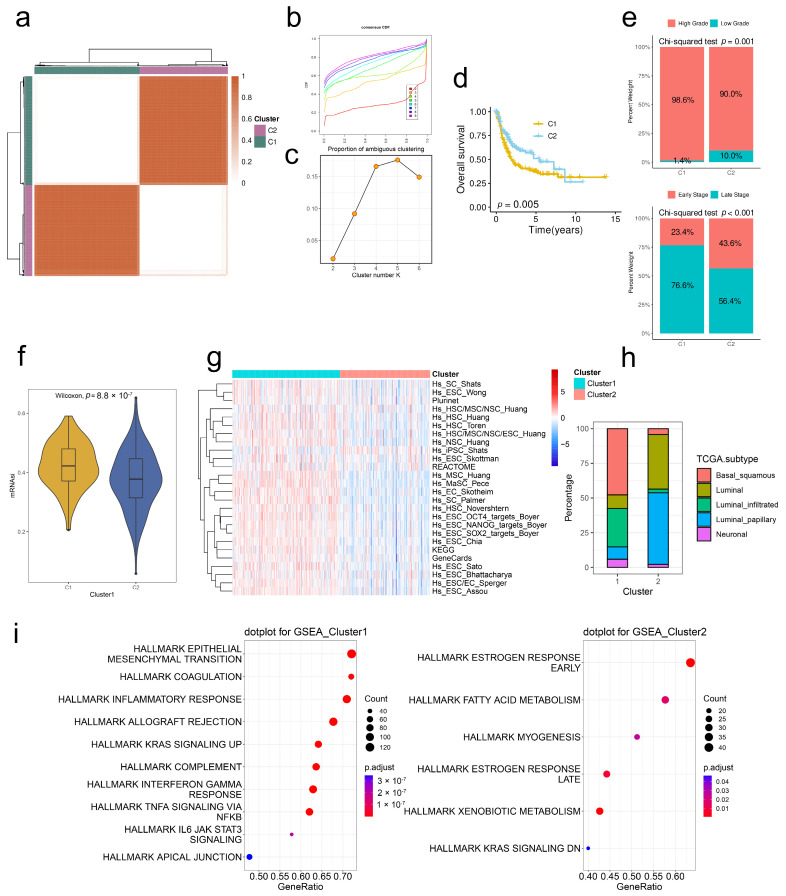
Stem. Sig. stratifies the TME of TCGA-BLCA into two molecular subclusters. (**a**) The consensus score matrix when k = 2. The CDF curves (**b**) and PAC scores (**c**) of the consensus matrix for each k. (**d**) KM curves with log-rank test revealed the poor prognosis of cluster 1. (**e**) Upper panel: The proportion of high- and low-grade between two clusters. Lower panel: The proportion of late- and early-stage between two clusters. Late stage, stages III and IV, early stage, stages I and II. (**f**) The distribution of mRNAsi between two clusters. (**g**) The heatmap showed the enrichment scores of 26 stemness-related signatures in two clusters. (**h**) The distribution of TCGA molecular subtypes between two clusters. (**i**) GSEA analysis demonstrated upregulated hallmark gene sets in two clusters.

**Figure 4 ijms-24-04760-f004:**
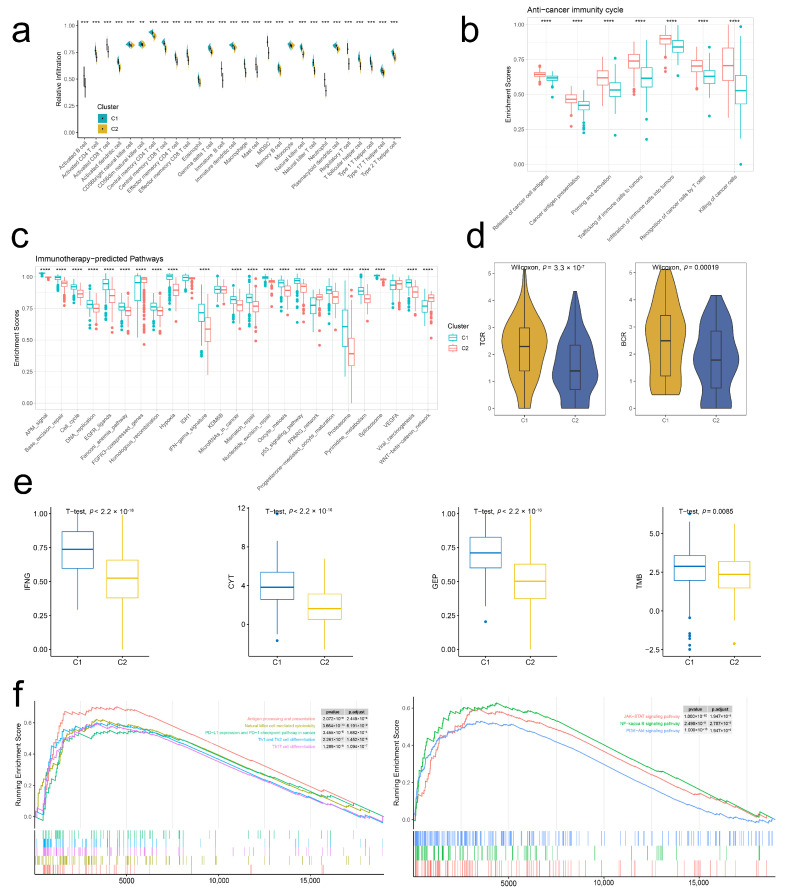
Stem. Sig. models an inflamed TME of TCGA-BLCA. (**a**) The infiltration abundances of 28 immune cell subsets between two clusters. (**b**) The distribution of the enrichment scores of a seven-step anti-tumor cycle between two clusters. (**c**) The distribution of the enrichment scores of immunotherapy-predicted pathways between two clusters. (**d**) The distribution of TCR and BCR evenness between two clusters. (**e**) The distribution of activities of four immunotherapy-predicted factors between two clusters. (**f**) Upregulated pathways in cluster 1. ** *p* <0.01, *** *p*<0.001, **** *p* <0.0001.

**Figure 5 ijms-24-04760-f005:**
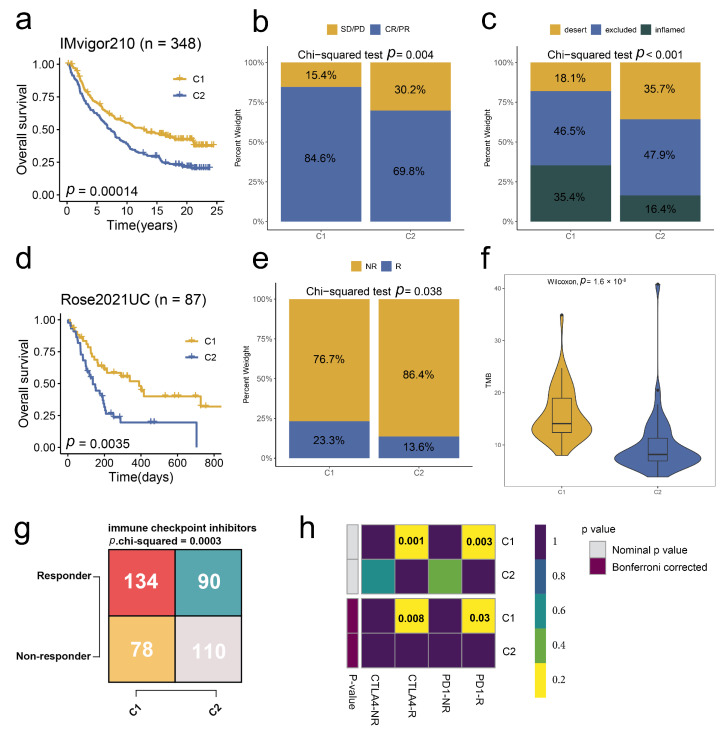
Stem. Sig. predicts immunotherapeutic responses in ICIs-treated BCa cohorts. (**a**) KM curves with log-rank test revealed the good prognosis of cluster 1. (**b**) The proportions of SD/PD and CR/PR between the two clusters. (**c**) The proportions of the immune phenotypes between two clusters. (**a**–**c**) Data were analyzed in the IMvigor210 dataset. (**d**) KM curves with log-rank test revealed the good prognosis of cluster 1. (**e**) The proportions of R and NR between the two clusters. (**f**) The distribution of TMB between two clusters. (**g**) A contingency table of the numbers of responders and non-responders to ICIs therapy between two clusters. Data were analyzed by the TIDE algorithm. (**h**) A contingency table between responses to ICIs therapy (anti-PD-1 and anti-CTLA-4) and two clusters based on the SubMap module of the GenePattern online framework.

**Figure 6 ijms-24-04760-f006:**
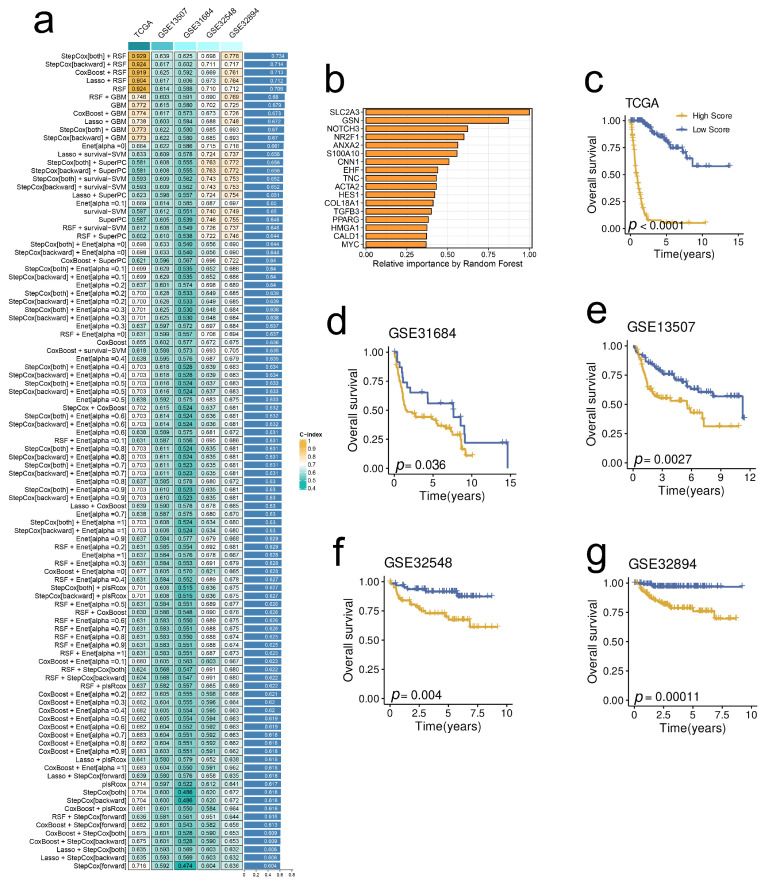
A prognostic model derived from the Stem. Sig. by an integrated machine-learning framework. (**a**) A total of 101 combinations of machine learning algorithms via a 10-fold cross-validation framework. The C-index of each model was calculated across validation datasets, including TCGA-BLCA, GSE13507, GSE31684, GSE32548, and GSE32894 datasets. (**b**) The importance of the 17 most valuable genes based on the RSF algorithm. Kaplan-Meier survival curve of OS between patients with a high score and with a low score in the TCGA-BLCA (**c**), GSE13507 (**d**), GSE31684 (**e**), GSE32548 (**f**), and GSE32894 (**g**) datasets.

**Figure 7 ijms-24-04760-f007:**
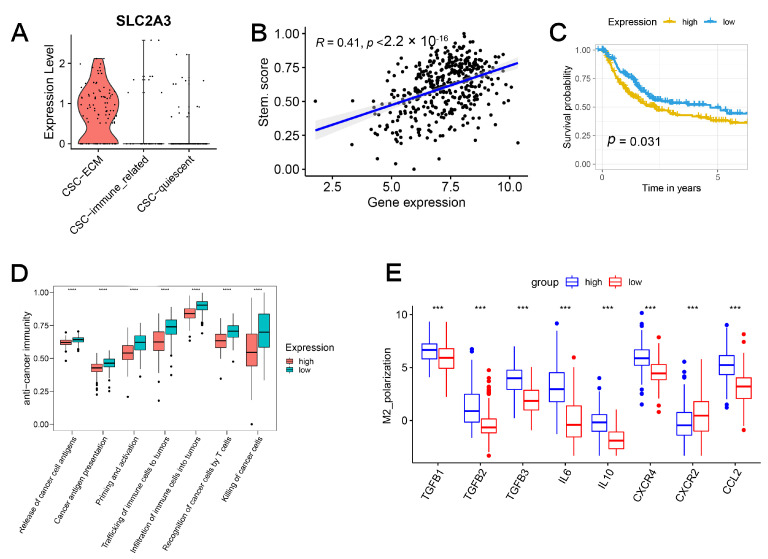
SLC2A3 is upregulated in ECM-related CSCs and related to impaired cancer-immunity cycle. (**A**) SLC2A3 was upregulated in ECM-related CSCs. (**B**) Positive correlation between the expression of SLC2A3 and stemness enrichment score. (**C**) Overexpressed SLC2A3 indicated a worse prognosis. (**D**) Overexpressed SLC2A3 indicated an impaired cancer-immunity cycle. (**E**) M2 macrophage polarization factors were upregulated in the high-expression group.

**Figure 8 ijms-24-04760-f008:**
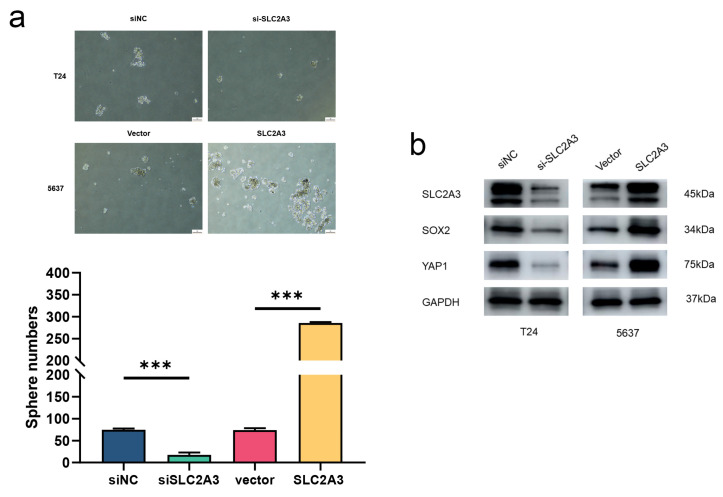
SLC2A3 overexpression promotes CSC traits, which can be suppressed by SLC2A3 inhibition. (**a**) Representative images of tumor spheres of indicated SLC2A3 expression. (**b**) Western blot was performed with the indicated antibodies. *** *p* < 0.001.

## Data Availability

All data used in this study are publicly available, as described in the Method section. The unique identifiers for public cohorts are described in the paper. The underlying code for this study is not publicly available but may be made available to qualified researchers upon reasonable request from the corresponding author.

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
