# Peer review of "Crosstalk between Mesenchymal Stem Cells and Cancer Stem Cells Reveals a Novel Stemness-Related Signature to Predict Prognosis and Immunotherapy Responses for Bladder Cancer Patients"

_ijms, 2023, doi:10.3390/ijms24054760_

Round 1

Reviewer 1 Report

The authors have to be praised for investigating the crosstalk between MSC and CSC in the bladder cancer setting.

The study resulted to be well written, extensively explained and clear. It could add significant information to the current knowledge and may be the ground for future studies about target therapy for bladder cancer.

Images need to be sharpened before publication. After this improvement and English spellcheck the manuscript will be suitable for publication. 

Author Response

Comment: Images need to be sharpened before publication. After this improvement and English spellcheck the manuscript will be suitable for publication.

Response: We thank the reviewer for pointing out this issue. And we have addressed these concerns in the revised manuscript.

Reviewer 2 Report

The authors found that deciphering crosstalk between MSCs and CSCs identified a Stem. Sig. that predicted prognosis and response to immunotherapy for BCa with the help of bioinformatics and a small number of validation experiments. The study design, conduct, results and conclusions are relatively reliable and integrated.

We have some comments that we would like to discuss with the authors.

1. What can be detected by single cell sequencing?

2.What are the possible biological roles of SLC2A3?

3.What experiments can be performed for the functional verification of SLC2A3? 

Author Response

Comment 1. What can be detected by single cell sequencing?

Response: Single-cell sequencing, including genomics, transcriptomics, epigenomics, proteomics, and metabolomics sequencing, is a powerful tool to decipher the cellular and molecular landscape at a single-cell resolution, unlike bulk sequencing, which provides averaged data. The use of single-cell sequencing in cancer research has revolutionized our understanding of the biological characteristics and dynamics within cancer lesions. In our study, we employed scRNA-seq to decipher the communication networks between MSCs and CSCs, construct gene regulatory networks, and uncover tumor heterogeneity.

Comment 2.What are the possible biological roles of SLC2A3?

Response: SLC2A3 (Solute Carrier Family 2 Member 3), also named GLUT3, is a protein-coding gene. The protein encoded enables dehydroascorbic acid transmembrane transporter activity, glucose binding activity, and glucose transmembrane transporter activity. Besides, as an integral component of the plasma membrane, it is involved in glucose import across the plasma membrane and transport across the blood-brain barrier. The biological roles of SLC2A3 in cancer have also been largely studied in the literature. Dai et al uncovered that AMPK/CREB1/GLUT3 axis may be a promising therapeutic target for colorectal cancer [1]. The implication of the Glut3-YAP-dependent Signaling Circuit in colorectal cancer has been reported in the study by Kuo et al. [2].

Refs:

[1]. Dai W, Xu Y, Mo S, et al. GLUT3 induced by AMPK/CREB1 axis is key for withstanding energy stress and augments the efficacy of current colorectal cancer therapies. Signal Transduct Target Ther. 2020;5(1):177. Published 2020 Sep 2. doi:10.1038/s41392-020-00220-9

[2]. Kuo CC, Ling HH, Chiang MC, et al. Metastatic Colorectal Cancer Rewrites Metabolic Program Through a Glut3-YAP-dependent Signaling Circuit. Theranostics. 2019;9(9):2526-2540. Published 2019 Apr 13. doi:10.7150/thno.32915

Comment 3.What experiments can be performed for the functional verification of SLC2A3?

Response: Since SLC2A3 is a protein-coding mRNA, a large number of assays can be employed for functional verification. qRT-PCR, immunoblotting, immunofluorescence, and Immunohistochemistry (IHC) can be performed to evaluate the expressions and subcellular locations of the SLC2A3 protein, as demonstrated in our study. Tumor sphere assays can be performed to determine the effect of SLC2A3 on tumor stemness. To further validate these findings, a series of in vivo assays can also be employed.

Reviewer 3 Report

Ma et. al. perform a comprehensive characterization of a subpopulation of cells identified in publicly available datasets. They then created a stem gene signature which lead to the validation of SCL2A3 as a driver of the phenotype. Although the works utilized solid methodology, I do have some concerns regarding the initial discovery datasets and how that might bias their findings, as well as the true nature of the gene

Overall, the manuscript is well constructed and is easy to follow. If the authors can address the below concerns, I feel that the paper is appropriate for this journal.

Major concerns:

1)    The single cell datasets that were merged for the initial discovery set were originally generated in very different ways. One was depleted of CD45+ cells prior to sequencing while the other was not. It would be helpful to see the 2 scRNAseq dataset clustered independently and cell types identified using a package such as singleR and then after the data are combined to see the relationship between the clusters and any batch effect.

2)    Can the authors expand upon what is known about MSC and CSC in the bladder and cite prior studies in which identified markers and their frequency within the overall tumor population. This would help to explain the rational for only using a very limited number of markers to identify the MSC and CSC populations. Also, are those cells at a frequency similar to what has been previously reported?

3)    The MSC cells in cluster 0 express UPK1B, which is a marker of urothelial cells. This would suggest that this cluster is composed, at least in part, of epithelial cells. Also, the authors label cluster 0 as immune related, yet one of the scRNA dataset is devoid of immune cells. Admittedly, the cell population could be expressing cell intrinsic immune-related genes, but also could be do to batch effect as pointed out in comment 1. Could the authors show the expression of a canonical set of bladder, MSC, and CSC specific genes for each cluster in both the initial clustering and cell specific clustering.

4)    Performing consensus clustering based on the stem sig. is not, as the authors described unsupervised. The analysis was supervised based on the stem sig. which was derived on a 2-group solution initially. Also, based on the genes within the signature, many are expressed within the tumor cells themselves so stating that the authors clustered the “TME” of TCGA tumors seems inaccurate.

5)    Based on the stem signature, clustering and immune genes, it seems the data simply recapitulate the basal and superficial cell types. Basal cells are thought of as the stem cell of the urothelium and its expression is related to the basal/squamous or more generally the basal subtype of MIBC. The Ba/Sq subtype has increased signatures for stemness as well as increased immune features, which is noted in their discussion. Could the author please show overlap between their groups and the various molecular subtypes of bladder cancer. Does the stem sig. correlate to the basal gene expression and/or outperform molecular subtype-based survival on their given validation datasets.

6)    The authors conclude that SCL2A3 is a key driver of stemness, however simply seed a plate and see how many spheres form. However, CSC should keep forming sphere and not/have a delayed exhaustion over time. Does over-expression or knockdown effect the ability of the cell lines to form spheres over multiple passages?

Minor concerns:

1)    Please add a color scale bar to all heatmaps

2)    The supplemental figures lack legends

3)    Resolution of the main text figures renders the images almost uninterpretable.

Author Response

Major concerns

Comment 1: The single cell datasets that were merged for the initial discovery set were originally generated in very different ways. One was depleted of CD45+ cells prior to sequencing while the other was not. It would be helpful to see the 2 scRNAseq dataset clustered independently and cell types identified using a package such as singleR and then after the data are combined to see the relationship between the clusters and any batch effect.

Response: We thank the reviewer for pointing out this issue.

(1). We re-analyzed the two scRNA-seq datasets and presented the whole analytic pipeline in Fig. S1. In brief, we followed the standard pipeline of Seurat and performed quality control, normalization, scaling, dimension reduction, and clustering in each dataset independently. SingleR package was employed to annotate these cell types and importantly, we have removed the NK cell population from the GSE130001 to eliminate the possible batch effect. An integrated scRNA-seq dataset was then obtained. Intriguingly, the number of clustering of our newly integrated scRNA-seq was consistent with the previous, and so was the cell type annotation by MSC and CSC markers. This result suggested, to some extent, the previously unremoved NK cells may not interfere with our findings.

(2). On the other hand, the annotation results by SingleR were consistent with one previous study [1], indicating the accuracy of the scRNA-seq processing pipeline.

(3). No obvious batch effect was observed in the merged dataset (Fig. S1I-J).

Ref:

[1] Yao J, Liu Y, Yang J, et al. Single-Cell Sequencing Reveals that DBI is the Key Gene and Potential Therapeutic Target in Quiescent Bladder Cancer Stem Cells. Front Genet. 2022;13:904536. Published 2022 Jun 3. doi:10.3389/fgene.2022.904536

Comment 2: Can the authors expand upon what is known about MSC and CSC in the bladder and cite prior studies in which identified markers and their frequency within the overall tumor population. This would help to explain the rational for only using a very limited number of markers to identify the MSC and CSC populations. Also, are those cells at a frequency similar to what has been previously reported ?

Response: We thank the reviewer for pointing out this issue.

(1). The up-to-date understanding of MSCs and CSCs in bladder cancer has been provided in the revised discussion section (about lines 230-248).

(2). We indeed understand your concerns about the accuracy of the identified MSCs and CSCs populations. Disappointingly, we failed to find the frequencies of MSCs and CSCs within the overall tumor population since a majority of relevant studies chose to report the total number of cells instead of the detailed number of MSCs and CSCs [1-2]. There are two main approaches for screening CSCs, by specific markers [1-3] or pseudotime analysis [4]. Wang et al. [1] identified bladder cancer stem cells based on CD44 and ALDH1A1. Lai et al. [2] found that one subpopulation was completely separated from other clusters in the UMAP plot (similar results were observed in our study) and had exclusively highly regulated CSC markers SOX9 and SOX2. These markers were comprehensively employed in our study to identify potential CSCs.

(3). These mentioned publications have been cited in the revised manuscript as refs 6, 9-11.

Refs:

[1]. Wang H, Mei Y, Luo C, et al. Single-Cell Analyses Reveal Mechanisms of Cancer Stem Cell Maintenance and Epithelial-Mesenchymal Transition in Recurrent Bladder Cancer. Clin Cancer Res. 2021;27(22):6265-6278. doi:10.1158/1078-0432.CCR-20-4796

[2]. Lai H, Cheng X, Liu Q, et al. Single-cell RNA sequencing reveals the epithelial cell heterogeneity and invasive subpopulation in human bladder cancer. Int J Cancer. 2021;149(12):2099-2115. doi:10.1002/ijc.33794

[3]. Fang D, Kitamura H. Cancer stem cells and epithelial-mesenchymal transition in urothelial carcinoma: Possible pathways and potential therapeutic approaches. Int J Urol. 2018;25(1):7-17. doi:10.1111/iju.13404

[4]. Wang L, He T, Liu J, et al. Pan-cancer analysis reveals tumor-associated macrophage communication in the tumor microenvironment. Exp Hematol Oncol. 2021;10(1):31. Published 2021 May 10. doi:10.1186/s40164-021-00226-1

Comment 3: The MSC cells in cluster 0 express UPK1B, which is a marker of urothelial cells. This would suggest that this cluster is composed, at least in part, of epithelial cells. Also, the authors label cluster 0 as immune related, yet one of the scRNA dataset is devoid of immune cells. Admittedly, the cell population could be expressing cell intrinsic immune-related genes, but also could be do to batch effect as pointed out in comment 1. Could the authors show the expression of a canonical set of bladder, MSC, and CSC specific genes for each cluster in both the initial clustering and cell specific clustering.

Response: We thank the reviewer for pointing out this issue.

(1). We analyzed the expression levels of UPK1B in the integrated scRNA-seq dataset and observed the exclusively upregulated expression in “epithelial cells” annotated by SingleR (Fig. to comment 3A) and in “other types” by manual annotation (Fig. to comment 3B). To further validate our findings, we also explored the TISCH2 online tool (http://tisch.comp-genomics.org/home/) to uncover the expression of UPK1B in the BLCA_GSE130001 dataset. Resultantly, UPK1B is overexpressed in epithelial cells, followed by endothelial cells, myofibroblasts, and fibroblasts (Fig. to comment 3C). The possible reason for UPK1B expressed in cluster 0 (also cluster 2) of MSC cells may be that the baseline of expression was low while cluster 1 of MSC had much lower levels compared to the other two clusters.

(2). The term “innate immune-related MSCs” just indicated the biological regulation of this MSC subtype in regulating innate immunity, instead of the intrinsic nature of immunity. The interactions between MSC and the immune system have been addressed in literature [1-2]. Besides, we have removed the immune cells subpopulation from GSE130001.

(3). We regret to inform you that there is no canonical set of the bladder, MSC, and CSC-specific genes reported in published literature. The markers of MSCs and CSCs in initial clustering have been illustrated in Fig. S1H. Fig. to comment 3D & E demonstrated the expression levels of each marker in each subpopulation of MSCs and CSCs.

Fig. to comment 3 (provided in the word file)

Refs:

[1]. Li N, Hua J. Interactions between mesenchymal stem cells and the immune system. Cell Mol Life Sci. 2017;74(13):2345-2360. doi:10.1007/s00018-017-2473-5

[2]. Jiang W, Xu J. Immune modulation by mesenchymal stem cells. Cell Prolif. 2020;53(1):e12712. doi:10.1111/cpr.12712

Comment 4: Performing consensus clustering based on the stem sig. is not, as the authors described unsupervised. The analysis was supervised based on the stem sig. which was derived on a 2-group solution initially. Also, based on the genes within the signature, many are expressed within the tumor cells themselves so stating that the authors clustered the “TME” of TCGA tumors seems inaccurate.

Response: We politely disagree.

(1). The clustering method employed in our study was implemented in the ConsensusClusterPlus R package. And this algorithm was used for determining cluster count and membership by stability evidence in unsupervised analysis, as stated in the formal documentation (https://www.bioconductor.org/packages/release/bioc/html/ConsensusClusterPlus.html). Consistently, previous studies also used this method to stratify tumor samples into several molecular subtypes based on the expression profiles of certain signature genes [1-2].

(2). We indeed got your concern about the accurate description of the term. Even though many of these genes of the signature were overexpressed in tumor cells, the signature itself can divide the TCGA-BLCA into two subtypes. Since the two subtypes demonstrated excellent discrepancies in immune cell subsets, we decided to use this term. Additionally, a similar description has been largely adopted in published studies [1-4].

Refs:

[1]. Wang L, He T, Liu J, et al. Pan-cancer analysis reveals tumor-associated macrophage communication in the tumor microenvironment. Exp Hematol Oncol. 2021;10(1):31. Published 2021 May 10. doi:10.1186/s40164-021-00226-1

[2]. Chen H, Yang W, Xue X, Li Y, Jin Z, Ji Z. Integrated Analysis Revealed an Inflammatory Cancer-Associated Fibroblast-Based Subtypes with Promising Implications in Predicting the Prognosis and Immunotherapeutic Response of Bladder Cancer Patients. Int J Mol Sci. 2022;23(24):15970. Published 2022 Dec 15. doi:10.3390/ijms232415970

[3]. Zheng H, Liu H, Li H, et al. Characterization of stem cell landscape and identification of stemness-relevant prognostic gene signature to aid immunotherapy in colorectal cancer. Stem Cell Res Ther. 2022;13(1):244. Published 2022 Jun 9. doi:10.1186/s13287-022-02913-0

[4]. Qing X, Xu W, Liu S, Chen Z, Ye C, Zhang Y. Molecular Characteristics, Clinical Significance, and Cancer Immune Interactions of Angiogenesis-Associated Genes in Gastric Cancer. Front Immunol. 2022;13:843077. Published 2022 Feb 22. doi:10.3389/fimmu.2022.843077

Comment 5: Based on the stem signature, clustering and immune genes, it seems the data simply recapitulate the basal and superficial cell types. Basal cells are thought of as the stem cell of the urothelium and its expression is related to the basal/squamous or more generally the basal subtype of MIBC. The Ba/Sq subtype has increased signatures for stemness as well as increased immune features, which is noted in their discussion. Could the author please show overlap between their groups and the various molecular subtypes of bladder cancer. Does the stem sig. correlate to the basal gene expression and/or outperform molecular subtype-based survival on their given validation datasets.

Response: We thank the reviewer for pointing out this issue.

(1). Robertson et al. [1] reported three main molecular subtypes of TCGA-BLCA, namely luminal subtypes (further divided into Luminal-papillary, Luminal-Infiltrated, and Luminal), one “Basal/Squamous” subtype, and one “Neuronal” subtype. Their classification system was based on Bayesian NMF with consensus hierarchical clustering of RNA-seq profiles. Similarly, our classification was also derived from unsupervised consensus clustering of the RNA-seq profiles of TCGA-BLCA.

(2). The distribution of previously proposed molecular subtypes between our two clusters has been provided in Fig. 3H. As expected, cluster 1 (high stemness) had a significantly higher proportion of Ba/Sq subtype compared to cluster 2.

(3). We extracted Ba/Sq subtype-specific markers from the original paper by Robertson et al. [1] and formed them into a signature named “Ba/Sq Sig.”. Pearson correlation analysis demonstrated a positive and significant correlation between Ba/Sq Sig. and Stem. Sig. (Cor = 0.26, P < 0.0001). On the other hand, the correlation of less than 0.3 partially indicated the intrinsic differences between two signatures at the mRNAs level (Fig. to comment 5).

(4). We politely decided not to analyze the survival discrepancies between our developed clusters and previously proposed molecular subtypes. We in the present study concentrated on the molecular subtypes derived from the Stem. Sig. by exploring the stemness level, survival, dysregulated pathways, and immunological landscapes. The performance in survival prediction or discrimination matters between cluster 1 and cluster 2 of our study, not with previously proposed molecular subtypes.

Fig. to comment 5 (provided in the word file)

Refs:

[1]. Robertson AG, Kim J, Al-Ahmadie H, Bellmunt J, Guo G, Cherniack AD, et al. Comprehensive Molecular Characterization of Muscle-Invasive Bladder Cancer. Cell 2017;171:540-556.e25. doi:10.1016/j.cell.2017.09.007.

Comment 6: The authors conclude that SCL2A3 is a key driver of stemness, however simply seed a plate and see how many spheres form. However, CSC should keep forming sphere and not/have a delayed exhaustion over time. Does over-expression or knockdown effect the ability of the cell lines to form spheres over multiple passages?

Response: We thank the reviewer for pointing out this issue.

(1). In fact, protein levels of cancer stemness markers and sphere formation capacity are two main approaches to investigating cancer stemness as well-documented in previous studies [1]. Zheng et al. [2] also explored the effect of COLEC12 on colorectal cancer stemness by these two assays. The present study mainly concentrated on decoding the stemness of TME based on Stem. Sig. and preliminarily identified candidate markers of which effects inferred the bladder cancer stemness. In-depth exploration of the biological functions of SCL2A3 is an exciting future area of investigation for our lab.

(2). As for the evaluation of the effect of SCL2A3 in spheres formation over multiple passages, we failed to find the rationality due to limited supporting evidence [1-2].

Refs:

[1]. Lin Yang, Jingya Sun, Meiqian Li, Yiming Long, Dianzheng Zhang, Hongqian Guo, Ruimin Huang, Jun Yan; Oxidized Low-Density Lipoprotein Links Hypercholesterolemia and Bladder Cancer Aggressiveness by Promoting Cancer Stemness. Cancer Res 15 November 2021; 81 (22): 5720–5732. doi:10.1158/0008-5472.CAN-21-0646

[2]. Zheng H, Liu H, Li H, et al. Characterization of stem cell landscape and identification of stemness-relevant prognostic gene signature to aid immunotherapy in colorectal cancer. Stem Cell Res Ther. 2022;13(1):244. Published 2022 Jun 9. doi:10.1186/s13287-022-02913-0

Minor concerns:

Comment 7 Please add a color scale bar to all heatmaps

Response: Thanks for your suggestion. And we added color scale bars to all heatmaps.

Comment 8 The supplemental figures lack legends

Response: Thanks for your suggestion. We arranged the manuscript based on the journal template. The supplemental figure legends have been placed under the Supplementary Materials section following 5. Conclusions Section.

Comment 9 Resolution of the main text figures renders the images almost uninterpretable.

Response: Thanks for your comment. The resolution of the main text figures was 300 dpi which met the requirement of the journal. Reviewers could zoom in to get a clearer vision.

Round 2

Reviewer 3 Report

I would like to thank the authors for their responses which addressed the concerns I raised regarding their manuscript. This study provides a nice base to build off of regarding the future understanding the signaling networks for MSCs and CSCs in bladder cancer.